# Assessment of Factors Influencing Personal Exposure to Air Pollution on Main Roads in Bogota: A Mixed-Method Study

**DOI:** 10.3390/medicina58081125

**Published:** 2022-08-19

**Authors:** Jeadran N. Malagón-Rojas, Eliana L. Parra-Barrera, Yesith Guillermo Toloza-Pérez, Hanna Soto, Luisa F. Lagos, Daniela Mendez, Andrea Rico, Julia Edith Almentero, Mónica A. Quintana-Cortes, Diana C. Pinzón-Silva, Andrés García, John A. Benavides-Piracón, Diana C. Zona-Rubio, Claudia Portilla, Maria A. Wilches-Mogollon, Sol A. Romero-Díaz, Luis Jorge Hernández-Florez, Ricardo Morales, Olga L. Sarmiento

**Affiliations:** 1Grupo de Salud Ambiental y Laboral, Instituto Nacional de Salud, Bogota 111321, Colombia; 2Maestría en Epidemiología, Facultad de Medicina, Universidad El Bosque, Bogota 110121, Colombia; 3Centro de Investigación en Ingeniería Ambiental, Universidad de los Andes, Bogota 111711, Colombia; 4Grupo de Cuidado Cardiorrespiratorio, Universidad Manuela Beltrán, Bogota 110121, Colombia; 5Grupo de Epidemiología, EPIANDES, Facultad de Medicina, Universidad de los Andes, Bogota 111711, Colombia; 6Grupo de Investigación Salud Pública, Educación y Profesionalismo, Universidad de los Andes, Bogotá 111711, Colombia

**Keywords:** air pollution, environmental exposures, public health, PM_2.5_, black carbon, active transport

## Abstract

*Background and Objectives*: Particulate Matter (PM), particles of variable but small diameter can penetrate the respiratory system via inhalation, causing respiratory and/or cardiovascular diseases. This study aims to evaluate the association of environmental particulate matter (PM_2.5_) and black carbon (BC) with respiratory health in users of different transport modes in four roads in Bogotá. *Materials and Methods*: this was a mixed-method study (including a cross sectional study and a qualitative description of the air quality perception), in 300 healthy participants, based on an exploratory sequential design. The respiratory effect was measured comparing the changes between pre- and post-spirometry. The PM_2.5_ and black carbon (BC) concentrations were measured using portable devices. Inhaled doses were also calculated for each participant according to the mode and route. Perception was approached through semi-structured interviews. The analysis included multivariate models and concurrent triangulation. *Results*: The concentration of matter and black carbon were greater in bus users (median 50.67 µg m^−3^; interquartile range (–IR): 306.7). We found greater inhaled dosages of air pollutants among bike users (16.41 µg m^−3^). We did not find changes in the spirometry parameter associated with air pollutants or transport modes. The participants reported a major sensory influence at the visual and olfactory level as perception of bad air quality. *Conclusions*: We observed greater inhaled doses among active transport users. Nevertheless, no pathological changes were identified in the spirometry parameters. People’s perceptions are a preponderant element in the assessment of air quality.

## 1. Introduction

The growth of cities has been accompanied by emerging effects such as mobility problems and exposure to environmental pollutants [1,2]. Studies have attempted the complex problem of the promotion of active transportation in cities and risks to human health through exposure to air pollution: exposure to ultrafine particulate matter in different transport modes has been associated with changes in heart rate [3], cardiovascular and respiratory pathologies [4,5,6,7,8,9], especially in people under 5 and over 65 years old [10,11].

Therefore, cities around the world have promoted walking and cycling or the combination of such methods with public transportation. This strategy promotes the reduction of emissions and improves air quality while promoting physical activity, particularly in citizens without time or resources to participate in sports or exercise [12,13,14]. Therefore, urban commuters are a group of concern, given their frequent proximity and repeated exposure to sources of combustion-related air pollutants [15,16]. Studies have estimated that commuting accounted for 30% and 32% of the inhaled dose of daily equivalent BC in Brisbane and Barcelona, respectively; similar results have been obtained for PM_2.5_ [17].

Bogotá is a capital city with more than nine million inhabitants. The vehicle fleet is composed of more than 2.4 million vehicles, mainly private (97%). The city has a complex public transport system which integrates bus rapid transit (BRT), bus, gondola lift services in the metropolitan area, and taxis. Most of the trips in the city are carried out using the public transport system (27%) [18].

It was estimated that approximately 21,000 deaths have been attributed to air pollution in the last 10 years in Bogotá [19]. The concentration of PM_2.5_ and BC in different transport microenvironments has been determined as up to six times higher in Rapid Transit public transport buses than for pedestrians and bicycle users in the same corridor [20]. However, the sample size and the spatial distribution of the few road segments sampled have limited the scope of those efforts to characterize personal exposure for commuters in Bogotá. In addition, transport microenvironment depends on other variables, such as sex, race, age, body mass index (BMI), lifestyle, usage of a face mask during the trip, history of respiratory diseases, smoking habits, condition of the highway and presence of green spaces [21]. Moreover, some authors have underscored the role of individual perception, travel attitudes and travel behavior as determinants of the transport microenvironment [22]. Therefore, qualitative studies could introduce additional elements for exposure analysis. Sensory perception, such as odor sensations, visually perceived changes in the environment and skin alterations, may be associated with higher pollution levels since these represent signs and symbols of exposure to polluted air [23].

In the era of active transport [24,25], it is important to evaluate the health risk of promoting physical activity in places where the concentration of PM_2.5_ is greater than the World Health Organization’s 2021 standards [26]. Strategic actions could reduce the adverse health impact of air pollution. Thus, this study aimed to evaluate the relationship between exposure to particulate matter, and black carbon, along with the perceptions and the respiratory health of users of modes of transport in prioritized routes in Bogotá, from the perspective of a mixed-method study.

## 2. Materials and Methods

We performed a mixed-method study with sequential design. Protocols followed were as those in [27] and following the recommendations of the STrengthening the Reporting of OBservational studies in Epidemiology (STROBE) checklist and the statement for observational studies and quality Standards for Reporting Qualitative Research (SRQR) results [28]. The description of the research process is described in Figure 1.

### 2.1. Study Area

Personal exposure to air pollutants and health variables measurements was assessed on four routes in urban areas of Bogotá (Figure 2). The routes were chosen based on these criteria: (i) routes located in areas with poor and medium air quality levels; (ii) road with high daily traffic by bicycle, minivan and regular buses of the Integrated Public Transport System (SITP); (iii) non previous monitored research; (iv) relevant routes for the Mobility and Environment City Major Bureau (Appendix A).

### 2.2. Sample and Sampling

The sample included volunteers from governmental agencies and universities, using a consecutive non-probability sampling. An expected occurrence of the event was estimated in 50% (*p* = 0.5), (*q* = 1−0.5 = 0.5). The precision of the estimate was ± 6% (*δ* = 0.06), and the confidence level was established to 95% (*α* = 0.05, *Zα* = 1.96). The minimum sample size was 267 participants.

For the qualitative component, the number of participants was considered according to the theoretical saturation criteria [29].

### 2.3. Data Collection

An online survey with individual information was completed by each participant between June 2019 and December 2020, including questions about perception of air quality, health and transportation. Volunteers were scheduled for monitoring on Thursday, Friday, and Sunday from 6 am to 10 am. Measurements were performed from the 19th of October 2020 to the 10th of November 2021. The participants were picked up from their home and driven to the initial point of the route in a minivan.

#### 2.3.1. Inhaled Doses

Inhaled doses were estimated using an Accelerometer (ActiGraph wGT3X-BT (Pensancola, FL, USA) [30]). The ActiGraph was attached to the participant’s waist at the beginning of the trip. For bicycle users, a second accelerometer was attached to the participant’s ankle. Data were recorded daily and downloaded using ActiLife^®^v5, licensed from Instituto Nacional de Salud (INS), Bogotá, Colombia. We estimated the metabolic equivalent of task (MET) using the Freedson equation and speed [31,32]. For bicycle users, we estimated the mean and median MET between the waist and ankle ActiGraph. The inhaled dose was estimated based on the procedure described by the EPA Handbook from the Environmental Protection Agency—EPA [33].

#### 2.3.2. Lung Function

Spirometry tests and analysis spirometry results were performed by two trained respiratory therapists with a Spirobank G (Medical International Research, Rome, Italy). Pre-exposure spirometry tests were performed before the trip and a second was performed two hours after the trip was completed. Each participant had at least three pre- and post-exposure spirometry tests.

The best spirometry tests (pre- and post-) were selected according to the acceptability, repeatability and interpretation criteria of the American Thoracic Society (ATS, New York, NY, USA) and European Respiratory Society [34]. FVC, FEV_1_, PEF and forced expiratory flow between 25% and 75% of the FVC (FEF_25–75%_) were measured. The FVC, FEV_1_, and PEF values corresponded to the percentage of predicted values computed according to the recommendation for the Hispanic population [35].

#### 2.3.3. Measurements for Monitoring Personal Exposure to PM_2.5_ and BC

PM_2.5_ concentrations were measured with a SidePakTM AM520, through a built-in sampling pump [36] and a DustTrak II Aerosol Monitor 8530 [37]. A flow calibration was performed before each use. BC was measured using a portable MicroAeth AE51 [38] that actively collects ambient air particles in the filter to determine the attenuation of the laser intensity through it [39].

Additionally, we included in the analysis the concentration of pollutants of anthropogenic and natural origin, reported by the stations of the Bogotá Air Quality Monitoring Network(–BAQMN), from at least two stations surrounding each route. We considered the hourly median, mean and standard deviation for PM_2.5_ data between 7 am and 11 am on the measurement days.

#### 2.3.4. Semi-Structured Interviews

A semi-structured interview guideline was designed to establish the knowledge, attitudes and practices related to the perception of air quality of participants who commute in a contaminated atmosphere. The social representation theory guided the structuring of the qualitative component (Table 1). Interview sessions were planned to last for 60 m and were conducted in Spanish, at the end of the trip.

### 2.4. Data Analysis

#### 2.4.1. Information Processing and Analysis—Quantitative Data

We constructed a database using each participant’s code as a key and linking sociodemographic, physical activity, spirometry, PM_2.5_, and BC variables. Data from air pollutants, GPS and the field registry were synchronized using Wolfram Mathematica^®^ software, V11.0, licensed from Universidad de Los Andes.

Means, medians, standard deviations and interquartile ranges were estimated for quantitative variables, and frequencies or percentages for qualitative variables. Pearson’s Chi-square test and Yates’s correction were used to compare the nominal or ordinal variables regarding sex, transport mode and route. Spearman correlation was used for quantitative variables. The level of statistical significance was *p* < 0.05. Data from measurements of personal exposure to PM_2.5_ were compared with data from the BAQMN to comprehensively characterize the PM_2.5_ concentrations.

A *Poisson* regression model was used to estimate the relative risk and to evaluate the differences between groups. We considered as dependent variable the percentage of change between the pre- and post-values in FVC, FEV_1_, PEF, FEF_25–75%_. The model considered the sex, route, median PM_2.5_ and BC concentrations, and transport mode as independent variables.

We estimated relative risks (RRs) with 95% confidence intervals. Analyses were performed in R version 4.0.2.

#### 2.4.2. Information Processing and Analysis—Qualitative Data

The analysis of the qualitative data was guided using strategies to ensure credibility, trustworthiness using software to ensure traceability of the transcription, coding process, and transferability (files were stored in MP4 format and on an external hard drive to ensure that all of the phases of analysis could be traced back to original interviews) [40].

Later, two of the researchers independently read the transcription to have a sense of immersion in the sensory experience of the subjects. Codification was established based on the theoretical elements raised and which refers to the knowledge, attitudes and practices that the participants have regarding air quality. The coding was conducted associating textual elements with theoretical constructs, generating a semantic network. The codes and emerged categories were validated by a third researcher. The findings were discussed by three of the co-authors until consensus on categories and subcategories was achieved. Finally, from the selective coding, a description of the content was made to determine the perceptions of the participants regarding the theoretical categories and to determine the relationships between the meanings of the categories. The scope of the analysis was limited to a descriptive level of content. The analysis of qualitative information was performed by ATLAS.ti Scientific Software Development GmbH (License from INS V.8).

#### 2.4.3. Triangulation and Integration of Results

The study included the integration of data through a joint display matrix to identify the relationships between the quantitative and qualitative datasets [41,42].

## 3. Results

### 3.1. Sociodemographic Characteristics

The study included 300 participants. Most of them were female (58.9%, *n* = 179) (Table 2). The average age was 31.61 ± 9.14 years. The average BMI was 23.04 ± 3.34. A total of 300 monitoring trips were completed; most of the trips were conducted in a bus (37.7%; *n* = 113), followed by minivan (35.3%; *n* = 106), and bicycle (27%; *n* = 81) (Table 2).

Participants mostly used private vehicles (41%) and buses (35.5%). Considering the transport mode, no relevant differences in age (Chi^2^ = 84.47; *p* = 0.149) or BMI (Chi^2^ = 357; *p* = 0.532) were observed. Bicycle users were mostly men (54%). The participants were formal workers (50%) and students (17.5%); the majority had a university degree (62%), knew how to ride a bicycle (90%), declared a good quality of life (96%) and were satisfied with their health (90%).

The number of trips was similar for each route: 116th Street (25.33%; *n*= 76); Southern Highway (24.7%; *n* = 74), Cali Avenue (25%; *n* = 75) and Quinto Centenario Avenue (25%; *n* = 75). The average duration of the route in minutes was 27.27 ± 9.4 for 116th Street, 29.37 ± 12.5 for Southern Highway, 31.4 ± 13.3 for Cali Avenue and 41.4 ± 16.5 for Quinto Centenario Avenue (Appendix A).

### 3.2. Air Quality Perception

Most of the participants declared that the quality of the air was regular or poor (71%) A link between air quality and health was also identified by 72% of the participants. According to the perception of the participants, the systems most frequently affected by air pollution were respiratory (94%), visual (72%), skin (67%), cardiovascular (41%) and gastrointestinal (17%). Air pollution was identified as the air became dark (75%). The most frequent protective element was a surgical/fabric face mask (40%) followed by N95 masks (7%). The participants mentioned practices such as closing windows (16%), opening windows (7%) and holding their breath (10%) during their daily travels.

### 3.3. Spirometry Parameters

Spirometry tests were performed on 300 participants. However, 13% (*n* = 39) of the spirometry tests were excluded because the criteria of the ATS were not fulfilled. All participants presented normal pre-and post-trip spirometry results, and alterations in lung volume were not evidenced (Appendix A). Change between the FEF_25–75_% pre- and post-trip volumes were observed in women (*p* = 0.04) but not in men (*p* = 0.12) (Appendix A). The FV_1_/FVC pre-and post-trip showed a significant difference (*p* = 0.03). This difference was more noticeable among women (*p* = 0.02), but it was also relevant among men (*p* = 0.04). However, these changes were not noticeable when the results were stratified by route or mode of transport (*p* > 0.05) (Table 2).

### 3.4. Personal Exposure

Non-normal distributions were observed for all modes and routes. The PM_2.5_ concentration was greatest in buses (median 50.67 µg m^−3^; IR: 306.7), followed by minivans (median 38.49 µg m^−3^; IR: 182.3) and bicycles (median 23.39 µg m^−3^; IR: 50.23). The differences were statistically significant (*p* < 0.05) (Appendix A). Similarly, BC concentrations were the highest in buses (median 29.94 µg m^−3^; IR: 116.3) and differed significantly from concentrations in bicycles (7.83 µg m^−3^; IR: 26.6) and minivan (18.54 µg m^−3^; IR: 68.6). The differences were statistically significant (*p* < 0.05) (Appendix A). The concentration of PM_2.5_ was significantly lower (*p* < 0.001) at 116th Street (median 15.66 µg m^−3^; IR: 59) than at the Southern Highway (median 60.18 µg m^−3^; IR: 202.7) and Cali Avenue (median 54.64 µg m^−3^; IR: 304.4). Similarly, the BC concentrations were greater at the Southern Highway (median 23.58 µg m^−3^; IR: 113.8) and Cali Avenue (median 22.21 µg m^-3^; IR: 102.8) than at 116th Street (median 6.37 µg m^−3^; IR: 30.9) (Appendix A). The concentrations of PM_2.5_ and BC were significantly lower on all routes and modes on Sundays than on weekdays (*p* < 0.001).

The background concentrations of PM_2.5_ reported by the BAQMN were greater at the Southern Highway and Cali Avenue. As expected, the PM_2.5_ concentrations were two or more times greater in the personal exposure measurements than those registered by the BAQMN (Appendix A).

### 3.5. Inhaled Dose

The average inhaled dose of PM_2.5_ was 11.50 µg ± 13.68, while the average inhaled dose of BC was 17.95 µg ± 23.36 (Appendix A). The inhaled doses of PM_2.5_ and BC were significantly higher among men than among women (*p* < 0.001) (Appendix A). The participants on bicycles experienced higher inhaled doses, followed by the participants on buses. Nevertheless, there was no association between inhaled doses of PM_2.5_ and BC and route or transport mode (*p* > 0.05). There was also no correlation between the spirometry parameters, age, body mass index and inhaled doses of PM_2.5_ and BC (*p* > 0.05). There was a positive correlation between the time of the commute and the inhaled doses of PM_2.5_ (Spearman 0.53; *p* < 0.001), and BC (Spearman 0.46; *p* < 0.001).

### 3.6. Multivariate Model

No risk factors were associated with changes in FVC, FEV_1_, PEF, or FEF_25–75%_ volumes (Appendix A).

### 3.7. Semi-Structured Interviews

A total of 44 semi structured interviews were conducted. Among the interviewees, 59% were women, 55% were between 18 and 26 years old, 43% traveled by private vehicle, 32% by bicycle and 25% by public transportation. Categories and subcategories are presented in the Appendix A.

#### 3.7.1. Knowledge of Air Quality and Its Relationship with Health

Knowledge was represented by the identification of contamination levels and the effects on health and quality of life. Air pollution levels were generally perceived as poor, regardless of their transport mode, the route on which they traveled, or their sex. Participants acknowledged being exposed to high levels of contamination. This perception was mediated by the sensory influence, with visual and olfactory perception being the two main referents of changes in environmental conditions. Thus, the smell of smoke and the visibility of smog were significant sensory evidence that the interviewees perceived when traveling through different areas of the city. Although many of the interviewees referred to this sensory perception generated by the exposure, the bicycle users recognized changes in these sensory experiences with greater intensity.


*P 8: male cent bici–8:2 The pollution is awful, and, well, in the quality of the air that one breathes. you perceive it in the smell. One says, not here, it smells terrible. Even if you are not seeing the smoke that is coming out of the car, you feel it.*


Public, cargo and private transportation, as well as industry, were recognized as the main sources of air pollution.


*P 6: male 116 van–6:6 The issue of mobile sources, especially public service, and cargo transportation, is what most influences (air quality), and I would say that the second activity would be industrial activities in some areas of the city.*


A link between the presence of symptoms and levels of exposure with both short- and long-term effects was identified by participants. Participants also declared that most of the effects on health were only observed in the long time.


*P26: male autosur bus–26:9 Allergies and discomforts such as eye and skin irritation, as in the short term some cancer, a disease in the background is long term because it results from being exposed for a long time.*


Participants underscored a relationship between air pollution and mental health. The perception of air quality arouses affective and emotional states reflected in changes in mood, such as feelings of anger, helplessness, irritation and stress, which can lead to alterations in the mental health of the participants.


*P 7: female cent bici–7:24 Clearly, between the perception of air quality, or between the perception of pollutants and emotional processes, there is something very, very strong, and that is that we read the world through what we perceive, and we are going to do that through our senses. Then, I do believe that there is a close relationship, I think that this fosters a lot of the mental health problems that we have today. I think that it also makes people prone to violence because it also becomes a violent, hostile context.*


#### 3.7.2. Attitudes toward Exposure

Attitudes towards changing behavior to avoid the risk were identified. One group was established by participants who had more empirical notions about how to protect themselves from air pollution and that showed more active attitudes toward the risk and assumed responsibility for their protection.

Active attitudes implied the use of elements to cover the nose and mouth in the face of direct exposure and the realization that physical activity reduces the risk of diseases. Additionally, changing attitudes implied modifying the routes used to travel from home to work. These attitudes were more frequently manifested by bicycle users, who showed a greater capability of influencing their exposure than minivan or bus users.


*P20: female cali van–20:8 when I see a car that is smoking too much, I always try to cover myself, not to inhale; that’s like my attitude. It’s about self-protection.*


A second group was constituted of participants with minor empirical knowledge of air pollution and risk. This group showed more passive behaviors, such as accepting the situation because it was not possible to influence the air pollution conditions. Attitudes of resignation were expressed, and naturalization was recognized as poor air quality and related to health risks. The participants’ attitudes suggested that being exposed was an unavoidable condition without a solution.


*P25: fem auto sur bus–25:10 It is difficult because what I tell you, it is like getting used to living in this atmosphere, like all the time for me it is normal.*


#### 3.7.3. Practices toward Exposure

We identified two main practices associated with air pollution exposure: individual (protection and reduction) and collective practices (policies and regulations, land use planning, information management) (Table 3). The first individual practice was the usage of personal protective equipment such as face masks or scarves. Related practices were rolling up/down windows, covering the mouth and nose, and holding the breath.

The second individual practice was the use of alternative modes of transport. The usage of bicycles for commuting is associated with a reduction in emissions and with a positive impact on individual health.

Collective practices were linked to “political actions” toward promoting health and environmental rights. It was widely recognized that there are not enough state actions to guarantee air quality in the city, but the participants also felt that there were not sufficient legal elements to allow them to participate in political decision-making related to air pollution. The participants expressed their impotence to influence political action related to air pollution and expressed a feeling of abandonment associated with the absence of a strong state that could protect citizens from private interests that can bend legislation on air pollution and sources of exposure.


*P 8: male cent bici–8:14. The muscle of the entities is deficient. Colombia is full of standards, at an environmental level, there are many standards, but the muscle to enforce those standards, the legal and economic muscle of the companies, can be more than the standard. So, if that does not change in this society (which is another of the things that bother me at a social level), substantial changes will not be seen.*


### 3.8. Triangulation of the Results

The results were consolidated in the joint display matrix shown in Table 3. A vast majority of the participants rated the air quality in the city as not good enough. Both the quantitative and qualitative data sets agreed that there was a link between health and air quality. From the qualitative perspective, air pollution was frequently associated with the presence of respiratory and visual discomfort and with symptoms such as anxiety, stress and irritation.

In addition, protective practices are considered an individual responsibility. The main action appears to be the use of face masks to reduce the exposure to air pollution. The qualitative analysis found other practices to reduce exposure to air pollution, such as changing the transport mode or the transport routes. In the qualitative analysis emerged a group of collective practices focused on political action. However, participants recognized that their capability to influence structural changes is limited by the social and economic context.

We did not find changes in the pre- and post-FVC, FEV_1_, PEF, or FEF_25–75%_ volumes. Multivariate models didn’t find associations between the spirometry parameters and concentrations of PM_2.5_, BC, routes and modes. Nevertheless, participants reported more frequently symptoms such as nose discharge and cough associated with air pollutants. Voluntaries did not considered respiratory effects related to the pulmonary volumes or capacity.

Participants on bicycles experienced higher inhaled doses. There was a positive correlation between the time of the commute and the inhaled doses of PM_2.5_ and BC. From a qualitative perspective, bike users tend to report that, in spite of being committed to the reduction of pollutant emissions, they considered themselves as the most exposed population to air contaminants.

## 4. Discussion

A mixed sequential explanatory study was conducted to evaluate the relationship between exposure to particulate matter PM_2.5_, BC and the respiratory health of users who commute by minivan, public transportation and bicycle on prioritized routes in Bogotá.

We found that air pollution perception was associated with the presence of respiratory discomfort and symptoms such as anxiety, stress, and irritation. These findings were like those previously reported by Buoli & Cols (2018), where the authors found that prolonged exposure to PM_2.5_ was associated with an increased risk of depressive symptoms [43]. Additionally, higher life satisfaction, more self-esteem and higher stress resilience are predicted by less air pollution (PM_10_) [44]. In addition, increase in particulate matter over an average PM_2.5_ concentration increases the likelihood of mental illness including depression, bronchial asthma, allergic respiratory diseases, and mortality in children [45,46].

Of the participants, 62% used some type of equipment to protect themselves from air pollution and this practice was more frequent among the bicycle users. These findings were in line with the individual practices declared in the interviews. The usage of face masks and practices such as closing windows to reduce the health consequences of air pollution have been reported [45]. Wearing a face mask appeared to effectively reduce symptoms and improve cardiovascular health measures in patients with coronary disease [47]. Masks also contribute to reducing the adverse effects of air pollution on blood pressure, even among healthy volunteers [48]. In addition, air pollution levels seemingly affect people’s behavior, reducing the practice of physical activity outdoors [49]. As air quality worsens, people tend to decrease their walking and cycling and to travel more by bus or subway [50].

There were no pathological changes in the spirometry, not even between sex, mode, or route. Several studies have found positive associations between physical activity and lung function levels in adults [51,52,53]. Most current evidence suggests that there is a positive association between active transport and physical activity. Even in cities with moderate air pollution levels, the benefits of physical activity outweigh the harm caused by air pollution [54]. Nevertheless, some authors have proposed that significant negative interaction effects exist between long-term exposure to PM_2.5_ and habitual physical activity, suggesting that the increased intake of PM_2.5_ due to physical activity may attenuate the benefits of habitual physical activity for lung function [55].

Although spirometry is the most common and practical test to measure lung function, it should be noted that it may fail to evaluate acute injury or inflammation from short-term exposure to contaminants. [56]. Some authors have proposed that FEV_1_ and FEF_25–75%_ flow are affected by short-time exposure to PM_2.5_, even in healthy participants [57,58,59]. On the other hand, it has been proposed that the short-term effects may be reflected through the IOS. The IOS is focused on evaluating the reactance and resistance of the air in the airway [60]. Some authors have proposed that elevated concentrations of PM_2.5_ and BC are associated with increase in the difference of the central and peripheral resistance to airflow [61].

The concentration of PM_2.5_ was highest in buses, followed by minivan and bicycles. Similarly, the concentrations of BC were highest in buses, and they differed significantly from the concentrations in bicycles and minivans. Our findings differ from the findings from Chaney et al. The authors studied the exposure to PM_2.5_ in different European cities. They found that the exposure rates were highest for cycling (18.0 μg/h) and walking (16.8 μg/h) and lowest for driving with windows closed (3.7 μg/h) [62]. The differences in the results may be explained by many factors. Traffic volume and diffusion conditions for the air pollutants may affect the exposure levels [63]. In our study, the bike lane was located on the pedestrian path in most of the measured tracks, which could reduce the exposure to PM_2.5_. Despite that there were not significant differences in the time of the journeys for each transport mode, bus and minivan users may remain motionless for longer, due to traffic jams or waiting in the bus station, while bike users are more likely to be in movement. Even when they may be more exposed to traffic exhausts, the exposure time may be minor compared to the time in the motorized modes.

Participants on bicycles showed greater PM_2.5_ and BC inhaled doses, followed by the participants on buses. Similarly, Peng reported that active commuters, were exposed to higher inhaled doses whereas car drivers inhaled the lowest doses of PM_2.5_ [64]. Driving with closed windows and air conditioning contributed to reducing the PM_2.5_ concentrations in minivans by 22% [64].

A comparative study that focused on comparing air pollution exposures in active vs. passive travel modes in European cities found that the estimated inhaled doses were greater for active modes (6.83 μg for walking and 2.78 μg for cycling) than for non-active modes (1.28 μg for light-rail trains, 1.24 μg for driving with the windows open, 1.23 μg for buses and 0.32 μg for driving with the windows closed). Bicycle users had higher rates of inhaled dose than commuters using automobiles or public transportation [65].

Despite these inhaled doses in active transport users, there was no correlation between the spirometry parameters, age, body mass index, and inhaled doses of PM_2.5_ and BC (*p* > 0.05). This may be explained by many factors. First, all the users wore a surgical facemask as a protection strategy against COVID-19. Many authors have shown that N95 and disposable facemasks may reduce the number of particles able to reach the alveoli to less than 2.5 microns [66]. Second is the physiological adjustment for individuals living at high altitude (2600 m.a.s.l.) [67]. Considering that only participants who have lived in Bogota for at least the last twelve months were included, it is expected that the respiratory frequency of the individuals in the study is adapted to the high-altitude conditions, showing lower inhaled doses of air pollutants, even among bike users. In this sense, the combination of facemask usage, physiological adaptation to altitude and sensitivity of the spirometry may explain the absence of associations between concentrations of PM_2.5_, BC, and respiratory outcomes.

Our study has many limitations. First, it is not possible to extrapolate the conclusion to other groups, or even other residents of the city, due to the nature and scope of cross-sectional studies. Second, despite trying to ensure a sample of healthy volunteers, we were unable to ensure that the participants had not been exposed to tobacco fumes or other pollutants the day before the measurements. Third, spirometry is not an easy test and may be affected by the participant. The repeated forced expiratory maneuvers may have annoyed the volunteers and reduced the quality of the obtained curves. Changes in the spirometry results may be associated with the nature of the exercise. We tried to control this potential bias by performing the second set of spirometry tests two hours after the tour ended. Fourth, accelerometry may underestimate MET inactive modes such as bicycling. We tried to control this potential bias by using an average between the MET obtained from the accelerometer on the hip and the ankle in the case of bicycle users. Fifth, although we tried to control exposure to air pollutants during the journey from home to the starting location, we could not ensure that exposure to PM_2.5_ and BC were equal to zero. Therefore, spirometry results could have been affected by exposure in the vehicle transporting the participants to the starting point. Sixth, exposure to PM_2.5_ and BC was measured with the SidePakTM AM520, the DustTrak II Aerosol Monitor 8530 and portable micro-ethalometers. A comparative analysis using gravimetric methods of sampling was not performed. To reduce the risk of bias in the PM_2.5_ and BC measurements, we used a protocol to calibrate the monitors in order to ensure the quality of the collected data. Seventh, since the study was performed on healthy participants, we were unable to ensure that the concentrations of PM_2.5_ and BC were “safe” enough to promote physical activity among high-risk populations (pregnant women, elderly individuals and children under the age of five). Eighth, the analysis of the transcripts from the semi-structured interviews was conducted by only one researcher, and only two coauthors validated the codes and categories obtained from the transcriptions. We tried to reduce this bias by the peer debriefing process. Finally, it was not possible to include the model or technology of the buses in the analysis. This may contribute to explaining better the differences in PM_2.5_ and BC concentrations among different routes.

## 5. Conclusions

The study shows relevant data for personal exposure in different transport modes through four main routes in Bogotá. We observed greater inhaled doses among bike users compared with bus and minivan participants. Despite that pathological changes in the spirometry parameters were not observed, the estimated concentration of PM_2.5_ and BC may have adverse health effects in the long term.

People’s perceptions are a preponderant element in the assessment of air quality, and it is suggested that environmental policy incorporates this type of information into its strategies. Incorporating individual and collective perception into actions implies optimizing information transfer and access processes in a way that promotes social learning and empowerment and therefore promoting the transformation of health and action through prevention measures.

This information is relevant since it can make citizens aware of the importance of air quality and therefore not only promote sustainable modes of transport but also stimulate citizen participation in activities for the formulation of public policies.

## Figures and Tables

**Figure 1 medicina-58-01125-f001:**
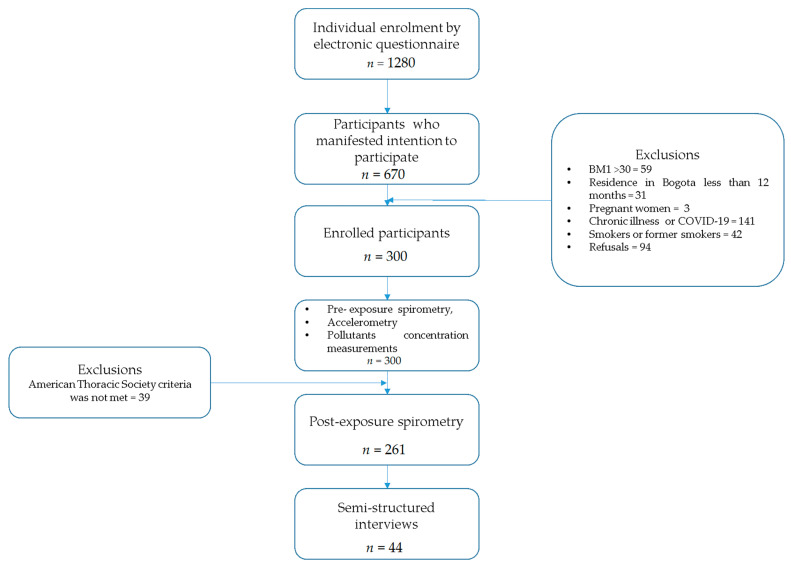
Flow chart description of the enrollment and research process.

**Figure 2 medicina-58-01125-f002:**
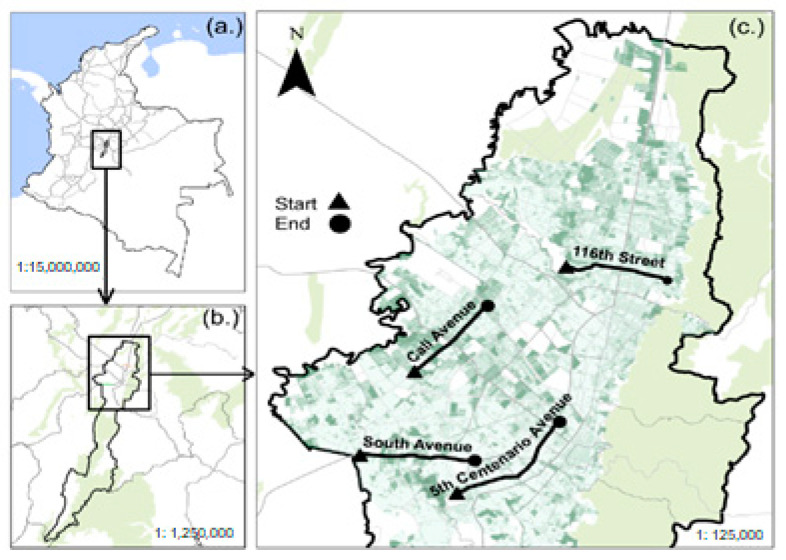
(**a**) Geographical location of Colombia and Bogotá. (**b**) Spatial location of Bogotá. (**c**) Locations of the four monitored routes. Triangles indicate the beginning of the routes, and circles indicate the end.

**Table 1 medicina-58-01125-t001:** Pre-established categories, definitions and questions from the semi-structured interview guide.

Category	Meaning	Questions
Knowledge of air quality and health	Mental representations preceding cognitive processes developed, socially constructed, and recreated during everyday interactions.	- What is your perception of the air quality in Bogotá? - What elements in the environment allow you to recognize the air pollution? - Do you know which pollutants affect air quality? - What do you consider to be the sources that impact air quality? - What do you consider to be the mode of transport with the higher exposure to pollutants? - Do you know if air quality has health effects? - What do you consider to be the effects on people’s physical, mental and emotional health and how are they evidenced? - Did you perceive the same air quality on the trip? Could you describe it?
Attitudes toward exposure	Means the orientation (positive or negative) before an object of the pre-existing social world. Cognitive, affective and behavioral components converge in them.	- Does pollution in the city alter your quality of life at all? - What attitude do you assume when exposed to a direct source of pollution? - Do you think you’ve gotten sick as a result of exposure to air quality pollutants? - Have you gotten sick? From what? Are you sick? - Did your attitude change from this episode?
Practices related to exposure	They express both the human experience and all those activities that materialize in a direct and daily relationship of people with the world. They are constituted as an action on reality.	- Do you consider that you take care of yourself and protect yourself against exposure to air pollutants? - Do you know about activities that are carried out in your environment to take care of your health in the face of exposure to air pollutants? Which one? - Of the activities mentioned, do you participate in any of them? - Does your perception of air quality interfere with decisions for your travel or the use of modes of transportation? - How do you think the city’s air quality could get better?

**Table 2 medicina-58-01125-t002:** Sociodemographic characteristics and spirometry parameters of participants.

Variable		Minivan	Bicycle	Bus	*p*
Age (years) (mean ± SD)		32.84 ± 10.35	31.75 ± 8.35	30.86 ± 8.78	>0.05
BMI (mean ± SD)		22.94 ± 3.46	22.95 ± 3.21	23.22 ± 3.36	>0.05
Sex (% (*n*))	Male	28.3% (*n* = 30)	59.26% (*n* = 48)	38.05% (*n* = 43)	>0.05
	Female	71.7% (*n* = 76)	40.74% (*n* = 33)	61.95% (*n* = 70)	
Routes (% (*n*))	Cali Avenue	23.58% (*n* = 25)	25.93% (*n* = 21)	25.66% (*n* = 29)	>0.05
	116th Street	26.42% (*n* = 28)	25.93% (*n* = 21)	23.89% (*n* = 27)	
	South Avenue	25.47% (*n* = 27)	22.22% (*n* = 18)	25.66% (*n* = 29)	
	Quinto Centenario Avenue	24.53% (*n* = 26)	25.93% (*n* = 21)	24.78% (*n* = 28)	
Spirometry parameters (mean ± SD)	Pre-FVC	3.79 ± 3.79	4.34 ± 4.34	3.87 ± 3.87	>0.05
	Post-FVC	3.72 ± 0.76	4.26 ± 0.86	3.88 ± 1.14	>0.05
	Pre-FEV_1_	3.08 ± 0.66	3.55 ± 0.68	3.21 ± 0.88	>0.05
	Post-FEV_1_	3.14 ± 0.62	3.53 ± 0.69	3.25 ± 0.85	>0.05
	Pre-FEF_25–75%_	3.48 ± 0.93	3.69 ± 1.02	3.47 ± 1.13	>0.05
	Post-FEF_25–75%_	3.58 ± 0.94	3.68 ± 1.02	3.63 ± 1.05	>0.05
	Pre-FEV_1_/FVC	82.86 ± 8.20	82.20 ± 5.70	83.40 ± 6.26	>0.05
	Post-FEV_1_/FVC	83.88 ± 9.47	82.92 ± 5.27	83.54 ± 9.82	>0.05

**Table 3 medicina-58-01125-t003:** Joint display of results.

Aim of the Study	Quantitative Mode	Qualitative Moment		Relationship	Summary of the Triangulation
	Summary of Results Related to the Objectives	Synthesis of Social Representations			
Describe the perception of the participants of air pollution in Bogotá.	Most of the participants reported that the quality of the air they breathe on their routes to places of work/study was not good enough (regular 47%, bad 24% and very bad 12%), with a link between air quality and health (72%). According to the perception of the participants, the human systems most frequently affected by air pollution were respiratory (94%), visual (72%), skin (67%), cardiovascular (41%), and gastrointestinal (17%). The participants stated that they use elements to protect against air pollution (62%). The most frequent element was surgical/cloth mask (40%), followed by N95 (7%). Likewise, the participants declared practices such as closing windows (16%); open windows (7%) and hold your breath (10%)”	Knowledge about air quality and its relationship with health.	The perception of air quality was mediated by the sensory experience (visual and olfactory) and the context where the sensation occurred. Recognizing an environment with high levels of contamination, public transport, cargo, and industries as the largest sources of contamination, as well as a high relationship between air pollution and health, which is expressed, on the one hand, in the appearance of respiratory diseases, eye irritation and skin discomfort, and on the other hand, the alteration of mental health expressed in changes in moods and stressful situations	Complementarity relationship between the quantitative-qualitative results.	The integrative analysis of the quantitative and qualitative results showed that the air quality in the city is poor, especially in routes by bus and in the Southern part of the city.
		Attitudes towards exposure	Two types of attitudes were identified, (1) associated with knowledge, mostly attitudes towards individual protection in order to reduce exposure. (2) attitudes of resignation, habits, naturalization where being exposed is an inevitable condition		
		Practices against exposure	Individual protection practices, mainly through the use of face masks became an element that establishes protection against contamination		
To estimate changes in lung volumes and respiratory symptoms in users according to the means of transport evaluated.	No differences were observed between pre and post spirometry within the FVC and FEV1 parameters.	Knowledge of air quality and its relationship with health	The respiratory system is recognized as the main affected system by air pollution. Symptoms such as fatigue, shortness of breath, cough, allergic reactions, sore throat, and nasal congestion were identified as manifestations of the contamination.	Neutrality relationship, since qualitative information does not provide results against spirometry alterations.	Social representations do not recognize spirometry alterations as a factor associated with contamination. However, a representation is constructed that relates pollution levels with health effects, especially with effects on the respiratory system.
		Attitudes towards exposure	The control over the potential exposure to air pollutants was assumed to be an individual responsibility The risk transferred to the person was maximized with the use of personal protection elements.		
		Exposure Practices	The influence of cognitive and affective factors was recognized as a determinant of environmental behaviour and its relationship with health care. This behaviour included individual actions to protect and implement healthy lifestyles		
To determine the concentration of P.M 2.5 and black carbon in the microenvironments evaluated.	The concentration of PM2.5 in buses was the highest (median 50.67 ug m^−3^; RI: 306.7), followed by cars (38.49; RI: 182.3), and bicycles. Similarly, the concentrations of CN in buses were the highest (29.94 ug m^−3^; RI: 116.3); these differed significantly from the concentrations in bicycles (7.83 ug m^−3^; RI: 26.6) and car (18.54 ug m^−3^; IR: 68.6)			Relation of complementarity, while the qualitative component allows expanding from the representations of the participants	The air pollutants concentration was variable according to route and mode. Perception of the air pollutants was not influenced by the transport mode or route
	The concentration of PM2.5 and BC as significantly lower in Street 116 compared to Southern avenue, and Cali Avenue.	Attitudes towards exposure	The use of protection elements is claimed, especially against direct and sudden exposure. Although some attitudes suggest a generalized awareness of the presence of air pollution, they also reflect a considerable level of apparent ambivalence, a naturalization that locates and signifies a distancing from the problem and a lack of social participation.		
	The mean inhaled dose of PM2.5 and BC doses were significantly higher among men compared to women. There was no association between inhaled PM2.5 and BC dose with route or mode of transport. There was no correlation between the spirometry parameters, age, body mass index, and inhaled dose of PM2.5 and carbon black. There was a positive correlation between travel time and inhaled dose of PM2.5 and BC.	Practices towards exposure	The use of the bicycle is claimed as the ideal means of transport, not only to reduce emissions, but also to improve and maintain optimal health conditions. They also suggest the adaptation of infrastructure and security conditions that allow this practice.		

## Data Availability

The study protocol and anonymized individual participant data that underlie the results reported in this manuscript, may be shared with investigators, whose proposed use of the data has been approved by the ethical and intellectual property committees of Instituto Nacional de Salud. Data can be provided for each individual participant, data meta-analysis, or other projects. The proposals should be directed to the corresponding author at Available online: jmalagon@ins.gov.co. To gain access, data requesters will need to sign a data access agreement, confirmed by J.N.M.-R. as principal investigator.

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
