# Peer review of "Assessment of Factors Influencing Personal Exposure to Air Pollution on Main Roads in Bogota: A Mixed-Method Study"

_medicina, 2022, doi:10.3390/medicina58081125_

Round 1

Reviewer 1 Report

The article is within the scope of the journal and presents satisfactory results for it to be accepted for publication. The results of table 2 seem to me that there are no significant differences between the samples. However, those who ride a bicycle are more exposed to air pollution than others (van and bus) and there is no approach in this text. I believe that your research was flawed in this sense, try to cite some references that emphasize this or help in the defense of your results. I think you need to improve the discussion about the effects of exposure to air pollution that people in this city are prone to. See a table with a summary of these likely consequences in this article: Santana, J.C.C.; Miranda, A.C.; Souza, L.; Yamamura, C.L.K.; Coelho, D.d.F.; Tambourgi, E.B.; Berssaneti, F.T.; Ho, L.L. clean Production of Biofuel from Waste Cooking Oil to Reduce Emissions, Fuel Cost, and Respiratory Disease Hospitalizations. Sustainability 2021, 13, 9185. https://doi.org/10.3390/su13169185 This will help improve your discussions and conclusions.

Author Response

Reviewer 1

Comment 1: The article is within the scope of the journal and presents satisfactory results for it to be accepted for publication. 

Response to the reviewer: many thanks for your valuable feedback.

Comment 2: The results of table 2 seem to me that there are no significant differences between the samples. However, those who ride a bicycle are more exposed to air pollution than others (van and bus) and there is no approach in this text. 

Response to the reviewer: we want to thank you for pointing this out. Indeed, the data presented in table 2 shows results which are not statically different. 

In addition, despite bike users seeming to be more exposed to air pollutants, the PM2.5 and BC were not significantly greater among these groups of participants. However, the inhaled doses of PM2.5 and BC were greater among bike users compared with bus and minivan users. It should be noticed that greater inhaled doses weren´t associated with changes in spirometry parameters.

To clarify the information to the readers, we have modified table 2 including a column with the p-value for each variable. Also, we have modified the numeral 3.5 as follows:

The average inhaled dose of PM2.5 was 11.50 µg ± 13.68, while the average inhaled dose of BC was 17.95 µg ± 23.36 (Supplementary table S2). The inhaled doses of PM2.5 and BC were significantly higher among men than among women (p < 0.001) (Supplementary table S4). Also, the participants on bicycles experienced higher inhaled doses, followed by the participants on buses. Nevertheless, there was no association between inhaled doses of PM2.5 and BC and route or transport mode (p > 0.05). Also, there was no correlation between the spirometry parameters, age, body mass index and inhaled doses of PM2.5 and BC (p > 0.05). There was a positive correlation between the time of the commute and the inhaled doses of PM2.5 (Spearman 0.53; p < 0.001), and BC (Spearman 0.46; p < 0.001). 

Comment 3: I believe that your research was flawed in this sense, try to cite some references that emphasize this or help in the defence of your results. I think you need to improve the discussion about the effects of exposure to air pollution that people in this city are prone to. See a table with a summary of these likely consequences in this article: Santana, J.C.C.; Miranda, A.C.; Souza, L.; Yamamura, C.L.K.; Coelho, D.d.F.; Tambourgi, E.B.; Berssaneti, F.T.; Ho, L.L. clean Production of Biofuel from Waste Cooking Oil to Reduce Emissions, Fuel Cost, and Respiratory Disease Hospitalizations. Sustainability 2021, 13, 9185. https://doi.org/10.3390/su13169185 This will help improve your discussions and conclusions.

Response to the reviewer: many thanks for your suggestion. We have read and included the suggested reference in the discussion section as follows:

A mixed sequential explanatory study was conducted to evaluate the relationship between exposure to particulate matter PM2.5, BC and the respiratory health of users who commute by minivan, public transportation and bicycle on prioritized routes in Bogotá.

We found that the air pollution perception was associated with the presence of respiratory discomfort and symptoms such as anxiety, stress, and irritation. These findings were like those previously reported by Buoli & Cols (2018), where the authors found that prolonged exposure to PM2.5 was associated with an increased risk of depressive symptoms [41]. Additionally, higher life satisfaction, more self-esteem and higher stress resilience are predicted by less air pollution (PM10) [42]. In addition, an increase in particulate matter over an average PM2.5 concentration, increases the likelihood of mental illness, including depression, bronchial asthma, allergic respiratory diseases, and mortality in children [43]. 

Of the participants, 62% used some type of equipment to protect themselves from air pollution and this practice was more frequent among bicycle users. These findings were in line with the individual practices declared in the interviews. The usage of face masks and practices such as closing windows to reduce the health consequences of air pollution has been reported [43]. Wearing a face mask appeared to effectively reduce symptoms and improve cardiovascular health measures in patients with coronary disease [44]. Also, masks contribute to reducing the adverse effects of air pollution on blood pressure, even among healthy volunteers [45]. In addition, air pollution levels seemingly affect people’s behaviour, reducing the practice of physical activity outdoors [46]. As air quality worsens, people tend to decrease their walking and cycling and travel more by bus or subway [47].

There were no pathological changes in the spirometry, not even between sex, mode, or route. Several studies have found positive associations between physical activity and lung function levels in adults [48–50]. Most current evidence suggests that there is a positive association between active transport and physical activity. Even in cities with moderate air pollution levels, the benefits of physical activity outweigh the harm caused by air pollution [51]. Nevertheless, some authors have proposed that significant negative interaction effects exist between long-term exposure to PM2.5 and habitual physical activity, suggesting that the increased intake of PM2.5 due to physical activity may attenuate the benefits of habitual physical activity for lung function [52]. 

Reviewer 2 Report

General comments

This manuscript is timely and well written which can add value to the scientific community and the study area if published after addressing the following minor points.

ü  English editions are required to be shown in each specific parts under

Abstract

ü  Highlight what you mean by mixed study in the context of your study.

ü  Line 23: …according to…to is missing

ü  Line 29-32: These statements are, I think irrelevant, better if you take line 29 to the introductory part, and put a recommending statement for the last one.

Introduction

ü  Well written, but it better if you add more on the industrialization and traffic conditions in Bogota so that the problem can be clearer.

ü  Please also consider the international situation in:

https://doi.org/10.1007/s11356-020-09132-1

Materials and Methods

ü  Well elaborated, don’t you think ethical clearance is required for the study?

Results

ü  Well presented, except some grammatical errors and unnecessary capitalization.

ü  Well triangulated

Discussion

ü  Well explained except Lines 393-396: They are introductory statements than result implication, better to avoid them.

Conclusion

ü  The conclusion is derived based on the results and discussion.

Author Response

Reviewer 2

General comments

This manuscript is timely and well written which can add value to the scientific community and the study area if published after addressing the following minor points.

Response to the reviewer: many thanks for your valuable feedback.

Abstract

Comment 1 Highlight what you mean by mixed study in the context of your study.

Response to the reviewer: many thanks for your suggestion. We have modified the abstract as follows:

Abstract: Particulate Matter (PM), particles of variable but small diameter could penetrate the respiratory system via inhalation, causing respiratory and/or cardiovascular diseases. This study aims to evaluate the association of environmental particulate matter (PM2.5) and black carbon (BC) with respiratory health in users of different transport modes on four roads in Bogotá. This was a mixed-method study (including a cross-sectional study and a qualitative description of the air quality perception), in 300 healthy participants, based on an exploratory sequential design. The respiratory effect was measured by comparing the changes between pre and post-spirometry. The PM2.5 and black carbon (BC) concentrations were measured using portable devices. Also, inhaled doses were calculated for each participant according to the mode and route. The perception was approached through semi-structured interviews. The analysis included multivariate models and concurrent triangulation. The concentration of matter and black carbon were greater in bus users (median 50.67 µg m-3; interquartile range –IR-: 306.7). We found greater inhaled dosages of air pollutants among bike users (16.41 µg m-3). We didn´t find changes in spirometry parameters associated with the air pollutants or transport modes. The participants reported a mayor sensory influence at the visual and olfactory level as the perception of bad air quality. We observed greater inhaled doses among active transport users. Nevertheless, we observed no pathological changes in the spirometry parameters. People’s perceptions are a preponderant element in the assessment of air quality.

Comment 2 Line 23: …according to…to is missing

Response to the reviewer: we have added the missed Word.

Comment 3 Line 29-32: These statements are, I think irrelevant, better if you take line 29 to the introductory part, and put a recommending statement for the last one.

Response: according to your suggestion we have taken out lines 29-32 and added a recommending statement to the final part.

Comment 4: Introduction

Well written, but it is better if you add more on the industrialization and traffic conditions in Bogota so that the problem can be clearer.

Response to the reviewer: many thanks for pointing this out. We have modified the introduction adding the required information as follows:

The growth of cities has been accompanied by emerging effects such as mobility problems and exposure to environmental pollutants [1,2]. Studies have attempted the complex problem of the promotion of active transportation in cities and risks to human health through exposure to air pollution: exposure to ultrafine particulate matter in different transport modes has been associated with changes in heart rate [3], cardiovascular, and respiratory pathologies [4–8], especially, in people under 5 and over 65 years old [9,10]. 

Therefore, cities around the world have promoted walking and cycling or combining such methods with public transportation. This strategy promotes the reduction of emissions and improves air quality while promoting physical activity, particularly for citizens without time or resources to participate in sports or exercise [11–13]. Therefore, urban commuters are a group of concern, given their frequent proximity and repeated exposure to sources of combustion-related air pollutants [14,15]. Studies have estimated that commuting accounted for 30% and 32% of the inhaled dose of daily equivalent BC in Brisbane and Barcelona, respectively; similar results have been obtained for PM2.5 [16].

Bogotá is a capital city with more than 9 million inhabitants. The vehicle fleet is composed of more than 2.4 million vehicles, mainly of them private (97%). The city counts on a complex public transport system which integrates bus rapid transit (BRT), bus, gondola lift services in the metropolitan area, and taxis. Most of the trips in the city are carried out using the public transport system (27%). 

Comment 5: Please also consider the international situation in: https://doi.org/10.1007/s11356-020-09132-1

Response to the reviewer: many thanks for your suggestion. We have included the reference in the introduction section.

Comment 6: Materials and Methods. Well elaborated, don’t you think ethical clearance is required for the study?

Response to the reviewer: We agree with the referee. Ethical considerations are required in this type of research. We have modified the institutional review board statement including the protocol approval act with the date as follows:

Institutional Review Board Statement: The study was conducted following the Declaration of Helsinki for studies involving humans. , and the protocol was approved by the Research Ethics and Methodologies Committee (CEMIN) at the National Institute of Health of Colombia (protocol code 014/2019, March 2019).

Comment 6: Results Well presented, except for some grammatical errors and unnecessary capitalization.

Response to the reviewer: we have checked the results section and we have changed it removing the unnecessary capitalization

Comment 7: Well triangulated

Response to the reviewer: Thank you very much for recognizing this aspect that we consider to be one of the strengths of the study.

Comment 8: Discussion. Well explained except Lines 393-396: They are introductory statements than result implication, better to avoid them.

Response to the reviewer: according to your suggestion we have deleted the introductory statements.

Comment 8: Conclusion. The conclusion is derived based on the results and discussion.

Response to the reviewer: many thanks for your valuable feedback.

This manuscript is a resubmission of an earlier submission. The following is a list of the peer review reports and author responses from that submission.

Round 1

Reviewer 1 Report

This study assessed the association of environmental PM2.5 and BC with respiratory health and physical activity in commuters traveling by different transport modes, based on the interviews and measurement of individual exposure to PM2.5 and BC. The topic is interesting and novelty, the questions of interview were established reasonable. However, the discussion on the exposure to PM and BC was not deepen. The manuscript also requires careful English editing, especially avoiding illogical expression. In general, the paper is suitable for publication after some major revision.

  1. The description about Materials and Methods was too long and too much detail. It is better to remove some of them into the supplementary and remain the significant part in the main text. For instance, the Data Collection in Section 2.3 could be summarized as one or two paragraph, but not split into several parts. And it is “2.3.5” in line 156, not “2.4.5”.
  2. In the results, it is better to plot more Figures in the main text, like comparing the air quality perception and the air pollutants concentration in line 231.
  3. In the section 3.5, the PM2.5and BC level in various street and transport modes were presented, therefore, it is necessary to further discuss the reason for these variation in the Discussion sector.
  4. Table 3 listed the relative risk of the participants in different routes, which is useful to discuss the health effect of air pollutants in various street. If the authors can combine the characteristics of road, proposing the countermeasures to reduce the relative risk in V Centenario Avenue, the result would be more comprehensive.
  5. In section 3.8-3.10, the authors described the individual perception, attitudes and practice  of air pollutants exposure based on the interview, which is reliable. But it is important to conclude the major factors of personal exposure.

6.In the discussion, lots of results from previous literature were cited. We hope to know more conclusion from your datasets. Moreover, the influence of PM2.5 was described in details, but lack of the description for BC.

  1. Line 400-404, It is better to discuss the reasons why the air pollutants leads to the perception changes here.
  2. Applying the superscript and subscript correctly, like PM5and PM10, µg m-3.
  3. Line 26: replace the 2.000 meters with 2,000 meters.
  4. Line 96: replace the 25.000 participants with 25,000 participants.

Reviewer 2 Report

Overall, the manuscript is characterized by the presence of a lot of information and statistical analyses not well linked to each other and it is difficult to understand the objective and the main message of the text.

I believe that a profound restructuring of the text and a thorough revision in terms of English language and clarity are necessary.

I suggest some specific revisions:

Title and aim should be aligned: the title refers to the factors influencing the exposure; the aim refers to the relationship between exposure and respiratory health and physical activity.

This discrepancy is reflected throughout the text, in the description of the main analyses, and in the choice of tables included in the main text and in the supplementary material.

Perhaps the authors should focus only on the innovative aspects (mixed-methods) used to analyze the exposure in different transport microenvironments. The relationship between exposure and health effect should be considered in a further manuscript. 

“Study area”: the authors, should better define criteria III and IV (it is not clear the meaning of the criterion from the point of view of the characterization of the roads). The authors should describe in the text the main characteristics of the four selected routes, according to the described criteria. In particular, in terms of traffic intensity, main or secondary road, air pollution levels, if one of the roads is considered as a “control” (i.e. low exposure)….

“Sample and sampling”: “the sample included 25.000 participants”. Please define better this aspect. Who these 25000 participants are?

In the sample size calculation, the authors refer to the “event”. Which event? Please, define it.

“Data collection”: “An online survey with individual information was completed by each participant between June 2019 and December 2020”. Do the authors refer to the 1280 subjects described in figure 1? Please, define it. Which kind of individual information was collected?

Please, describe the online survey in more detail. Who filled in the online survey? How were the subjects contacted? What kind of selection criteria were used?

Describe in the text the exclusion criteria and the reasons for their choice.

Furthermore, the authors should describe how it was decided which of the four roads the 300 participants should cross. Was it a random choice? Or were the participants used to crossing those streets during their normal life? how was the choice of the means of transport made (bicycle, bus, minivan)? on the basis of personal habits?

“Data analysis”: please describe in detail the Poisson model. The authors reported that ”For associations between sociodemographic factors and spirometry changes a Poisson model was completed”.  What analyses are the authors referring to? How did you compute the spirometry changes? For which variables were the analyses adjusted? Please, describe the analyses used to assess the relationship between spirometry and the exposure (type of route, mode of transport).

Qualitative data: The authors reported that “The scope of the analysis was limited to a descriptive level of content”. But, one of the main aspects of the mixed methods is the integration of the data: “explicit interrelating of the quantitative and qualitative component in a mixed methods study” (Plano Clark and Ivankova, 2015). The authors should describe how qualitative and quantitative phases, results, and data were integrated (Pluye et al., 2018). For instance, how data gathered by both research methods were brought together to form a complete picture (e.g., joint displays) and when integration occurred (e.g., during the data collection analysis or/and during the interpretation of qualitative and quantitative results). It is not sufficient a “descriptive level content” to define a mixed method.

It is not clear which information is added using qualitative data. How this information can complement the assessment of exposure? Please, define it more clearly.

On the other side, the authors reported that “The study included the integration of data through a triangulation matrix to identify the relationships”. But this aspect is not detailed described and the results are neither described nor discussed and they were reported only in Supplementary materials. 

Results, paragraph 3.4: one of the aims was the relationship between exposure and physical activity. Did the authors perform these analyses?

Results, paragraph 3.6: How did the authors define the “inhaled dose”? this is not described in the methods section.

Table 3: the title states “Post FEF25-75”, but the authors described the results as changes in FEF25-75. Please, define this aspect better.

Table 3. Results of the analyses on BC are missing.

Table 3: to better understand the meaning of these analyses, a more detailed description of the 3 roads is necessary, as explained above.

Supplementary materials: please include in the text the tables to which the authors refer (es. Supplementary Table 1). Many tables were included in the supplementary materials and it may be difficult to identify the correct ones. Moreover, the tables are not self-explanatory; please add some notes to define the acronyms and write more detailed titles.

Minor revisions:

Line 46: “in the last 10 years” instead of “in the last 10 years old”.

Lines 52-53: “In addition….depends…” the sentence is not clear.

Line 76: “on four routes of urban areas” instead of “on four routes urban areas”.

Lines 76-77: based on “the following” criteria.

Figure 1: please check the number; 670 participants + 648 subjects with exclusion criteria= 1318 (not 1280). Or are the exclusion criteria not mutually exclusive? Please, add the numbers in the second box of exclusion criteria.

“The criterious ATA”? or “The ATS/ERS criterion”?

Line 129: “interviews”. Which kind? Semi-structured?

Lines 218-222: Do these data come from the semistructured interview?

Results, line 214: The authors reported that 148 subjects were female. But, in table 2 the number of females is 179. Please check.

Table 2: please, add the statistical significance (i.e. p-value)  of the analyses.

Line 233: “Also, the link between air quality and health”. The sentence is not clear.

Line 247: “FEV1/FVC” instead of “FE/FVC”.

Line 248: “results showed a significant difference”. Increase? Decrease?

Line 292: “FEV1” instead of “VCF1”.

Reviewer 3 Report

In the manuscript ID: ijerph-1655330, entitled “Assessment of factors influencing personal exposure to air pollution on main roads in Bogota: a mixed-method study”, the authors aim to evaluate the relationship between exposure to PM2.5, BC, and the respiratory health of commuters in Bogotà. In general, I believe that the manuscript is really difficult to follow, many of its concepts are not clearly and logically explained. In particular, the study design is not very clear: I suggest that the authors revise the paragraph on the methods used in an important way. In addition, I do not understand if the study design allows such statistical robustness to the data as to justify the results. More detailed comments are reported below.

  • Abstract: I suggest that the authors report the main findings of the study here.
  • Abstract: The authors report that “ […] this is the first mixed-methods study focused on PM2.5, BC, and respiratory health effects in a city above 2.000 meters above sea level.”: what does it mean for the scientific community? what is it useful for?
  • Lines 33-36: The connection between the two reported sentences is not clear.
  • Line 77: Did the authors define a priori “zones with a wide range of air pollution levels”? How di they define “wide range”? Similarly, how is “high daily traffic” defined?
  • Figure 1: Are the sums correct? In addition, I suggest the authors to be consistent in all the info reported in the figure.
  • Line 265: Differences verified by which test?